# Conservative Tryptophan Residue in the Vicinity of an Active Site of the M15 Family l,d-Peptidases: A Key Element in the Catalysis

**DOI:** 10.3390/ijms241713249

**Published:** 2023-08-26

**Authors:** Galina V. Mikoulinskaia, Dmitry A. Prokhorov, Sergei V. Chernyshov, Daria S. Sitnikova, Arina G. Arakelian, Vladimir N. Uversky

**Affiliations:** 1Branch of Shemyakin & Ovchinnikov’s Institute of Bioorganic Chemistry, RAS, Prospekt Nauki, 6, 142290 Pushchino, Moscow Region, Russia; svch2@rambler.ru (S.V.C.); darya_sitnikova_2000@mail.ru (D.S.S.); arkatsachndir@gmail.com (A.G.A.); 2Institute of Theoretical and Experimental Biophysics, RAS, Institutskaya ul., 3, 142290 Pushchino, Moscow Region, Russia; dmitry_pr@rambler.ru; 3Laboratory of New Methods in Biology, Institute for Biological Instrumentation, Russian Academy of Sciences, Federal Research Center “Pushchino Scientific Center for Biological Research of the Russian Academy of Sciences”, 142290 Pushchino, Moscow Region, Russia; 4Department of Molecular Medicine, Morsani College of Medicine, University of South Florida, Tampa, FL 33612, USA; 5USF Health Byrd Alzheimer’s Research Institute, Morsani College of Medicine, University of South Florida, Tampa, FL 33612, USA

**Keywords:** l,d-peptidase, bacteriophage, catalysis, substrate binding, calcium

## Abstract

Bioinformatics analysis of the sequences of orthologous zinc-containing peptidases of the M15_C subfamily revealed the presence of a conserved tryptophan residue near the active site, which is not involved in the formation of the protein core. Site-directed mutagenesis of this Trp114/109 residue using two representatives of the family, l-alanoyl-d-glutamate peptidases of bacteriophages T5 (calcium-activated EndoT5) and RB49 (EndoRB49, without ion regulation) as examples, and further analysis of the ^1^H NMR spectra of the mutants showed that a decrease in the volume of the W → F → A residue leads to changes in the hydrophobic core and active center of the protein, and also decreases the affinity for regulatory Ca^2+^ in the EndoT5 mutants. The inactive T5W114A mutant lacks the ability to bind the substrate. In general, the conserved Trp114/109 residue, due to the spatial restrictions of its side chain, significantly affects the formation of the catalytically active form of the enzyme and is critical for catalysis.

## 1. Introduction

Bacteriophage L,D-peptidases are lytic enzymes (endolysins) that ensure the release of phage progeny from the host bacterium cell by hydrolyzing the peptide bond between L-alanine and D-glutamic acid residues of peptidoglycan, the main component of the bacterial cell wall. Since the l-Ala-d-Glu bond is always present in the type A peptidoglycan, which is characteristic of all Gram-negative and most Gram-positive bacteria [1], potentially, L-alanoyl-D-glutamate peptidases serve as the broad-spectrum antimicrobial agents (“enzybiotics”). Currently, several modular (having a domain structure of the molecule) L,D-peptidases have been identified [2,3,4], and several globular (single-domain) L,D-peptidases are known as well [5,6]. All of them belong to the M15_C family of zinc-containing peptidases.

Endolysins of bacteriophages infecting Gram-positive bacteria usually have a modular structure and have one or more N-terminal catalytic domains (CD) and one or more substrate binding C-terminal domains (SDs) [7]. SD determines bacteriospecificity by recognizing and non-covalently binding to a ligand located on the surface of a bacterial cell. Some SDs are structurally and functionally well characterized, such as LysM, which binds the GLcNAc of the peptidoglycan sugar backbone [8], Cpl-1, which binds teichoic acid choline residues [9], and SH3b, which binds the pentaglycine cross-bridges of staphylococcal peptidoglycan [10]. The peptidoglycan-binding domains of modular endolysins often contain tryptophan residues [11]. The interaction between the binding domain and the ligand is characterized by high affinity. A striking example of this is the PlyP35 SD of the *Listeria* phage, where the equilibrium affinity constant is in the pico- and nanomolar range, which is comparable or even exceeds the affinity of the antibody to the antigen [12]. It is believed that, due to its high affinity, SD keeps the enzyme firmly bound to the remnants of the cell wall of the host bacterium after phagolysis, preventing the diffusion of the active enzyme and the subsequent destruction of surrounding bacterial cells that are not yet infected with the phage and thereby serve as potential future hosts for progeny phage particles. Therefore, SD limits the processivity of an enzyme with a modular structure.

The outer membrane of Gram-negative bacteria prevents the very possibility of destruction of potential host cells by endolysin, limiting access to PG from outside. This is probably why phage endolysins that infect Gram-negative bacteria are, as a rule, small globular proteins with a single domain that combines the functions of catalysis and substrate binding. To date, there are no data on the mechanisms of substrate binding by globular endolysins. In the only study on substrate binding by phage endolysins of the Gram-negative *Pseudomonas* bacterium, the objects of study, KZ144 and EL188, turned out to be modular enzymes of chimeric origin. At the N-terminus of these proteins, there is an analogue of SD of modular phage endolysins of Gram-positive bacteria, which provides them with affinity for peptidoglycan of the A1 chemotype [13]. Therefore, the mechanisms of simultaneous substrate binding and catalysis by globular endolysins in general and L,D-peptidases in particular remain unexplored, primarily due to the limited structural data on these enzymes and the complexity of the peptidoglycan substrate structure.

In orthologous globular L,D-peptidases of bacteriophages T5 (Siphoviridae), RB43, and RB49 (pseudo T-even Myoviridae), a conserved tryptophan residue W114/110/109 was found, located near the active center and not directly involved in the formation of the hydrophobic core of the protein globule, being exposed to solution. Interestingly, in EndoT5, W114 is located in the mobile loop that binds the regulatory calcium ion [14], in contrast to EndoRB43 and EndoRB49, which lack calcium regulation. It was shown that the proximity of the conserved Trp114 residue to the active site of EndoT5 is a factor that allows the regulatory Ca^2+^ to have a stabilizing effect on the structure of the protein globule [15].

The goal of this work was to determine the roles of this conserved tryptophan residue W114/110/109 in the interaction of the L,D-peptidases with the substrate and its effects on the catalytic activity of these enzymes. To this end, we carried out site-directed mutagenesis of the conservative W114 of L,D-peptidase EndoT5 (with calcium activation) and W109 of L,D-peptidase EndoRB49 similar to it (without calcium activation) and analyzed changes in the structure and properties of enzymes induced by such mutations.

## 2. Results and Discussion

### 2.1. Bioinformatic Analysis of Phage l,d-Peptidases of the M15_C Family

The subfamily of zinc-containing M15_C peptidases, despite the limited number of biochemically characterized enzymes (all listed in the Section 1), has more than 850 potential members in the MEROPS database (https://www.ebi.ac.uk/merops/cgi-bin/famwrap/famcards/aln/m15c_xal.htm, accessed on 1 August 2023 [16]). Therefore, the structure-function relationships found for single proteins can be extrapolated to a large group. We have identified over a hundred protein sequences by searching for homologous L-alanoyl-D-glutamate peptylases in the taxon of tailed phages Caudovirales (taxid: 28883) using the PSI-BLAST program [17]. From the set of found homologues, 22 sequences were selected with sequence identities ranging from 36% to 80%. Figure 1 shows the multiple alignment of these sequences, a consensus sequence, and the result of secondary structure prediction.

Comparison of these results with the data on the three-dimensional structure established for EndoT5 [14] allows us to conclude that the main part of conserved amino acid residues is located within the α-helices and β-folds of the hydrophobic core. Strictly conservative residues, which are marked in Figure 1 by stars, are two histidine residues and aspartate coordinating zinc (in EndoT5 these are H66, D73, H133, and in EndoRB49 H62, D69, H117), as well as catalytic aspartate D130/D114. It should be noted that tryptophan 114/109 located near the active center in the unstructured region (marked with a red arrow in Figure 1) is strictly conserved in all representatives analyzed in this study, regardless of the degree of their general similarity. The position of this Trp residue indicates that it does not have the structure-forming function characteristic of other hydrophobic consensus amino acids, but may play an important role in the formation of the catalytically active conformation of the active site.

### 2.2. Site-Directed Mutagenesis and Mutant Activity Assay

To analyze the role of conserved Trp114/109 in catalysis using site-directed mutagenesis, we obtained EndoT5 and EndoRB49 mutants containing point substitutions: EndoT5W114A, EndoT5W114F, EndoRB49W109A, and EndoRB49W109F. The substitutions did not affect the solubility of the recombinant proteins. The study of the lytic activity of pure enzyme preparations showed that the alanine mutant EndoT5W114A was completely devoid of activity, while EndoRB49W109A retained residual activity at a level of 5% of the activity of the natural protein (Table 1). However, substitutions of tryptophan for functionally similar phenylalanine in both EndoT5 and EndoRB49 retained 30–40% of the activity of the native proteins (Table 1). It should be noted that the measurement of the lytic activity of EndoT5W114F required the addition of 100 μM CaCl_2_ to the reaction medium. At the same time, native EndoT5 does not need exogenous calcium to manifest its activity: a regulatory ion contained in the substrate, natural cell walls, is sufficient. Apparently, the need for exogenous Ca^2+^ is a consequence of a decrease in the affinity of EndoT5W114F for this regulatory ion.

### 2.3. Structural Basis of Functional Changes in EndoRB49(Zn^2+^) and EndoT5(Zn^2+^Ca^2+^) Mutants

Assignments of the Zn^2+^ and Zn^2+^Ca^2+^ forms of ^13^C^15^N-labelled EndoT5 were performed using the standard sets of 2D and 3D experiments and deposited to PDB (PDB ID 2MXZ and 8P3A, correspondently). The assignment of signals of the histidine residues of the Zn^2+^ form of orthologous EndoRB49 was performed using 1D-NOE with unlabeled protein and ^15^N HSQC experiments with ^13^C^15^N-labelled protein published in [16]. Based on these data, we analyzed mutation-induced structural changes in the ^1^H NMR spectra.

Analysis of the upfield part of the ^1^H NMR spectra of EndoT5(Zn^2+^Ca^2+^) and EndoRB49(Zn^2+^) forms of mutants showed that as the volume of the W → F → A substituted residue decreases, changes in the packing of the hydrophobic core of the globule are observed (Figure 2). In the case of EndoT5(Zn^2+^Ca^2+^), this is evidenced by the change in the position of the upfield resonance at −0.66 ppm from γ1H V93, which is part of one of the α-helices forming the hydrophobic core of the globule (Figure 2A). The changes are local in nature and do not affect the environment of L28, which is also part of the hydrophobic core; the position of its resonance at 0.683 ppm remains virtually unchanged. In the case of EndoRB49(Zn^2+^) forms of mutants, the upfield resonances of 0.193 ppm from γ2H and δ1H I88 (a residue similar to V93 of EndoT5) and −0.306 ppm from βH of neighboring A89 sequentially change in a similar way (Figure 2B). The shift in the positions of upfield resonances of these residues towards low fields, in the direction of the positions of the signals of free residues, indicates a consistent decrease in their fixation in the hydrophobic cores of enzymes.

Analysis of the amide part of the ^1^H NMR spectra of EndoT5(Zn^2+^Ca^2+^) and EndoRB49(Zn^2+^) forms of the mutants showed that as the volume of the W → F → A residue decreases, structural changes are observed in the active site of the enzymes, which are symbate with the changes described above in the hydrophobic core (Figure 3). A marker of this in the ^1^H spectra of EndoT5 mutants is a gradual shift of the position of the 15.70 ppm ε2HN histidine signal of the H133 of the active center, which coordinates the catalytic Zn^2+^, to higher fields (Figure 3A), which correlates with a decrease in activity to its complete loss by the alanine mutant EndoT5W114A. Similar changes are also characteristic of the EndoRB49 mutants: a downfield shift of the 14.11 ppm signal of the ε2HN H117 to higher fields reflects a change in the mutual configuration of the H117 and H62 active-site histidine residues coordinating the catalytic Zn^2+^ (Figure 3B). The change in the position of the resonance at 14.11 ppm also correlates with a decrease in specific activity in the series EndoRB49wt → EndoRB49W109F → EndoRB49W109A. Thus, substitutions of the conserved Trp114/109 destabilize the active site of the studied peptidases, causing disruption of the conjugation of zinc-coordinating amino acid residues, which inevitably increases the activation barrier.

### 2.4. Effect of W114 Substitutions on the Ability of EndoT5 to Bind Regulatory Calcium

The replacement of the conservative tryptophan of EndoT5 with phenylalanine causes a decrease in the affinity for the regulatory calcium ion: activity, as already mentioned above, requires a molar excess of the exogenously added Ca^2+^ ion. The alanine mutant is completely inactive even in the presence of 100 μM CaCl_2_.

Structurally, a decrease in the affinity for regulatory calcium (W > F > A) is a decrease in the intensity of the upfield γ2H I82 signal at −0.291 ppm (Figure 2A). This signal occurs in native EndoT5 during the formation of a catalytically active open protein conformation [14]. In the three-dimensional structure of the catalytically active Zn^2+^/Ca^2+^ form of EndoT5 (PDB 8P3A), which we established, the Ca^2+^-binding loop is fixed on the globular domain. The shift of the γ2H I82 signal is caused by a change in the degree of mutual fixation of the side chains of the exposed residues I82 and W84 to the surface, reflecting the spatial position of the loop. The loss of intensity of the upfield signal of γ2H Ile82 at −0.291 ppm, accompanied by its shift (−0.081 ppm) to low fields, indicates a decrease in the population of molecules in the active conformation (Figure 2A). For the EndoT5W114F mutant, there is a 63% reduction in the signal intensity at −0.291 ppm. Therefore, the population of molecules in the open conformation is 37%. In the limiting case of EndoT5W114A, the active conformation of the enzyme is not determined at all.

The response of EndoT5 mutants to the presence of Ca^2+^ ions can be observed by changing the signal intensity of amide protons G118 at 10.73 ppm and ε2HN H121 at 10.48 ppm (Figure 3A). These residues are part of the EndoT5 calcium-binding loop, which, as mentioned above, is fixed on the globular domain in the structure of the native PDB 8P3A protein. Loop fixation is mediated by Ca^2+^ binding and exposure of the W114 residue to the surface, which leads to the “opening” of the active site and the acquisition of a catalytically active conformation by the protein. When the W114 residue is replaced by F, the calcium-binding capacity decreases, and the bound Ca^2+^ does not lead to reliable fixation of the loop, reducing the proportion of molecules in the active “open” conformation. The alanine residue at position 114 deprives the loop of steric fixation, and the enzyme of the catalytically active conformation. The cooperativity of structural changes in the molecule mediated by the binding of the Ca^2+^ ion is lost. Thus, the tryptophan residue, not directly involved in the coordination of the regulatory calcium ion, plays an important role in the stabilization of the protein globule in the active conformation.

### 2.5. Effect of the W114A Substitution on the Ability of the Enzyme to Bind the Substrate

The complete absence of intrinsic activity in the EndoT5W114A mutant made it possible to analyze its relative ability to bind the substrate. The experiment was carried out under conditions of competition for the substrate of a molar excess of the mutant protein with 1 nM native EndoT5, whose activity in the absence of other potential competitor proteins was taken as 100%. Bovine serum albumin BSA served as a negative control: having no affinity for bacterial cell wall peptidoglycan, it is unable to compete for the substrate with EndoT5 and does not affect its activity (Figure 4). The catalytic aspartate mutant EndoT5D130A, also completely inactive, was used as a positive binding control.

Figure 4 shows that the activity of EndoT5 drops in the presence of a 30- and 300-fold molar excess of EndoT5D130A to 85% and 38% of the maximum, respectively. A 1200-fold molar excess of EndoT5D130A completely inhibits EndoT5 activity. The decrease in the lytic activity of the native protein can be explained by competition for the substrate with the D130A mutant for binding sites. Therefore, the catalytic aspartate mutant, despite the loss of activity, retained the ability to bind the substrate.

The W114A mutant in 30- and 300-fold excess is not able to compete for the substrate with the native protein: there is not a decrease, but even a slight increase in the lytic activity of EndoT5. A slight increase in activity can be caused by single acts of binding-dissociation of the mutant W114A with the substrate, which makes the substrate slightly more accessible to hydrolysis by the active enzyme. A 1200-fold molar excess of W114A led to an insignificant decrease in the enzymatic activity of native EndoT5 up to 60% of the maximum. Therefore, the ability of the EndoT5W114A mutant to compete with EndoT5 is sharply reduced compared to the catalytic aspartate mutant. It is possible that the decrease in the ability to bind the substrate is mediated by a change in the conformation of the protein globule as a whole, but it cannot be ruled out that the conserved tryptophan residue is directly involved in hydrophobic interactions with the peptidoglycan substrate.

### 2.6. Intrinsic Disorder Predispositions of the Wild Type and Mutant Proteins

To get some additional clues on the effects of the analyzed amino acid substitutions, intrinsic disorder predispositions of the wild type proteins and tryptophan mutants were compared. Figure 5A represents the overlaid intrinsic disorder profiles generated for EndoT5wt, EndoT5W114F, and EndoT5W114A by one of the more accurate per-residue disorder predictors, PONDR^®^ VSL2 [18]. Corresponding data for EndoRB49wt, EndoRB49W109F, and EndoRB49W109A are shown in Figure 5B. This analysis revealed that mutations noticeably affect the local disorder propensity of these proteins, which increases in the order of W < F < A.

In EndoT5wt, mutations at position 114 increased local disorder propensity from 0.2990 in the wild type protein to 0.3649 and 0.4860 in W114F and W114A mutants, respectively. Similarly, in EndoRB49, local disorder propensity at position 109 increases from 0.3037 in the wild type protein to 0.3583 and 0.4539 in F and A mutants. It is likely that such mutation-induced increase in the local disorder propensity contributes to the structural perturbations caused by the substitutions of the conserved W to F and A.

To illustrate the scale of the variability of local disorder propensity in the vicinity of the conserved tryptophan residues, Figure 6 represents aligned disorder profiles of the 22 members of the phage L,D-peptidases of the M15_C family. Analysis of data shown in this figure shows that despite noticeable variability in the details of the per-residue disorder predisposition throughout their sequences, intrinsic disorder profiles of these proteins show remarkable overall similarity. The least disorder variability is observed at two regions—in the vicinity of residue 70 and 118 of the aligned profiles, with position 118 corresponding to the conservative tryptophan residues analyzed in this study. Interestingly, that position 70 corresponds to the aspartate residue of the active site involved in the coordination of catalytic zinc (see Figure 1). According to this analysis, local disorder at position 118 ranges from 0.1959 to 0.4321, with the mean disorder score being 0.302 ± 0.063.

This rather strong conservation of the peculiarities of disorder propensity distribution in the vicinity of the tryptophan of interest suggests that this region is of functional importance not only in EndoT5 and EndoRB49, but in other members of this protein family as well.

## 3. Materials and Methods

### 3.1. Bioinformatics Methods

The PSI-BLAST program [17] was used to search for homologous L-alanoyl-D-glutamate peptidases in the tailed phage taxon Caudovirales (taxid: 28883). Multiple alignment of amino acid sequences was performed using the online version of the Clustal W program (https://www.genome.jp/tools-bin/clustalw, accessed on 1 March 2023 [19]). Consensus sequences were searched using ESPript 3.0 [20]. The secondary structure was predicted using the Jpred 4 program [21].

### 3.2. Site-Directed Mutagenesis

Mutagenesis was carried out by the QuikChange method using high-precision thermostable polymerase Q5 (New England BioLabs, Ipswich, MA, USA). The temperature and time parameters during the polymerase chain reaction were selected taking into account the recommendations of the QuikChange protocol (Stratagene, San Diego, CA, USA) and NEB Tm-Calculator. The pEndoT5 and pEndoRB49 plasmids obtained by us earlier [6] were used as initial template DNA. The following primer pairs were used to replace tryptophan (TGG) with alanine (GCG):

T5WAup: 5′-CTTCGTTTTGGTGCTGATGCGAATGCTTCGGGAGACTATC-3′

T5WAlo: 5′-GATAGTCTCCCGAAGCATTCGCATCAGCACCAAAACGAAG-3′

RB49WAup: 5′-GTTGAATGGGGTGGGGATGCGACTAGTTTTAAAGATGGTCCTC-3′

RB49WAlo: 5′-GAGGACCATCTTTAAAACTAGTCGCATCCCCACCCCATTCAAC-3′

The following primer pairs were used to replace tryptophan (TGG) with phenylalanine (TTC):

T5WFup: 5′-CTTCGTTTTGGTGCTGATTTCAATGCTTCGGGAGACTATCAC-3′

T5WFlo: 5′-GTGATAGTCTCCCGAAGCATTGAAATCAGCACCAAAACGAAG-3′

RB49WFup: 5′-GTTGAATGGGGTGGGGATTTCACTAGTTTTAAAGATGGTCCTCACT-3′

RB49WFlo: 5′-AGTGAGGACCATCTTTAAAACTAGTGAAATCCCCACCCCATTCAAC-3′

### 3.3. Enzyme Preparations

Homogeneous preparations of EndoT5, EndoRB49, and mutant proteins EndoT5W114A, EndoT5W114F, EndoRB49W109A, EndoRB49W109F were excluded to electrophoretic homogeneity from the cells of producer strains using the laboratory procedure based on chromatographic techniques and described earlier [6]. The protein concentration in homogeneous preparations was determined spectrophotometrically from the absorbance at 280 nm, based on the molar extinction coefficient of the target protein. The protein concentration used in the acquisition of ^1^H-NMR 1D spectra was 0.8 mM.

### 3.4. Enzyme Activity Assay

Enzyme activity was determined by a spectrophotometric method based on a decrease in the optical density of a suspension of *E. coli* B cells preliminary permeabilized with chloroform as described [6]. An activity unit was defined as the quantity of enzyme that provides the rate of optical density decrease of 1.0 optical unit per minute. All activity data were calculated from at least 3 independent measurements.

### 3.5. NMR Spectroscopy

Chromatographically pure protein preparations were dialyzed three times against 100× volume of ammonia water (pH 9.3). The desalted aqueous protein solution was transferred into a glass flask, frozen in liquid nitrogen at −196 °C, and freeze-dried in an Inei-4 sublimation unit (Russia) at a residual pressure of ~60 Pa for ~8 h until the sample temperature equalized with room temperature. The dried protein was dissolved in 1.0 mL of a buffer of 50 mM CD_3_COOD, 3 mM Zn(NO_3_)_2_, 0.03% NaN_3_ (pH = 4.1) containing 8 M urea. To obtain Zn^2+^-containing forms, the preparations were dialyzed three times against 200 mL of a buffer of 50 mM CD_3_COOD, 3 mM Zn(NO_3_)_2_, 0.03% NaN_3_ (pH = 4.1). To obtain the two-ion (Zn^2+^/Ca^2+^) form of EndoT5 and its mutants, CaCl_2_ solution was added to the preparation to a final concentration of 3 mM. Subsequent renaturing dialysis was performed three times against 200 mL of a buffer of 50 mM CD_3_COOD, 3 mM Zn(NO_3_)_2_, 3 mM CaCl_2_, 0.03% NaN_3_ (pH = 4.1). Before measurements, 50 µL of D_2_O was added to each sample.

The ^1^H-NMR spectra were measured on an AVANCE-III 600 spectrometer (Bruker, Berlin, Germany), with an operating frequency on protons of 600 MHz, a spectral width of 14,383 Hz, a relaxation delay of 2 s, and a 90-degree pulse of 10 μs. To achieve a good signal-to-noise ratio, 1024 accumulations were sufficient. The sample temperature was 298 K.

## 4. Conclusions

Globular single-domain endolysins of bacteriophages infecting Gram-negative bacteria are active and stable antibacterial protein agents that are promising for use both in the treatment of bacterial infections (including antibiotic-resistant ones) and for non-therapeutic purposes for the needs of biotechnology. Functional analysis of the role of the conserved tryptophan of L,D-peptidases of the M15 family by site-directed mutagenesis is the first attempt to shed light on the mechanisms of binding and hydrolysis of the peptidoglycan substrate by single-domain globular endolysin.

Bioinformatics analysis of the sequences of orthologous zinc-containing L,D-peptidases of the M15_C subfamily revealed the presence of a conserved tryptophan residue near the active site, which is not involved in the formation of the protein core. Site-directed mutagenesis of the conservative W114/109 of two members of the L,D-peptidases EndoT5 (with calcium activation) and EndoRB49 (without it) showed changes in the structure and properties of the mutant proteins. Tryptophan substitutions for functionally similar phenylalanine in both EndoT5 and EndoRB49 retained one third of the lytic activity of natural proteins, EndoRB49W109A had a residual 5% activity, and EndoT5W114A was completely inactive. An analysis of one-dimensional ^1^H NMR spectra of EndoRB49(Zn^2+^) and EndoT5(Zn^2+^Ca^2+^) forms of mutants showed that a decrease in the volume of the W → F → A residue leads to changes in the hydrophobic core and active center: evidence of the first is a change in the position of the upfield resonance from γ1H V93 (EndoT5) or γ2,δ1H I88,βH A89 (EndoRB49), which is part of one of the α-helices of the hydrophobic core; the marker of the second is a shift in the position of the ε2HN signal of active center H117/H133 into the upfield region. In EndoT5 mutants, the affinity for the regulatory Ca^2+^ ion decreases, the structural marker of which is a decrease in the intensity of upfield signals from γ2H I82 and the downfield signals of amide protons of G118 and ε2HN of H121 in ^1^H NMR spectra (W > F > A), which reflects a deterioration in the fixation of the regulatory loop on the globular domain. The ability of the inactive mutant EndoT5W114A to compete with EndoT5 for the substrate peptidoglycan is sharply reduced, indicating the possible involvement of the conserved tryptophan residue in hydrophobic interactions with the substrate. Conservative tryptophan residues analyzed in this study are one of the two sites with the least disorder variability in L-alanoyl-D-glutamate peptidases of the M15_C family.

In summary, the conserved Trp114/109 residue indirectly influences the formation of the hydrophobic globule of both proteins, and in EndoT5, it also affects the fixation of the regulatory calcium-binding loop, ensuring the formation of a catalytically active “open” conformation of the protein. The strict conservation of the tryptophan residue in this position suggests the universality of the mechanism of functioning of all L-alanoyl-D-glutamate peptidases of the M15_C family, which degrade peptidoglycan and are widespread in tailed phages.

## Figures and Tables

**Figure 1 ijms-24-13249-f001:**
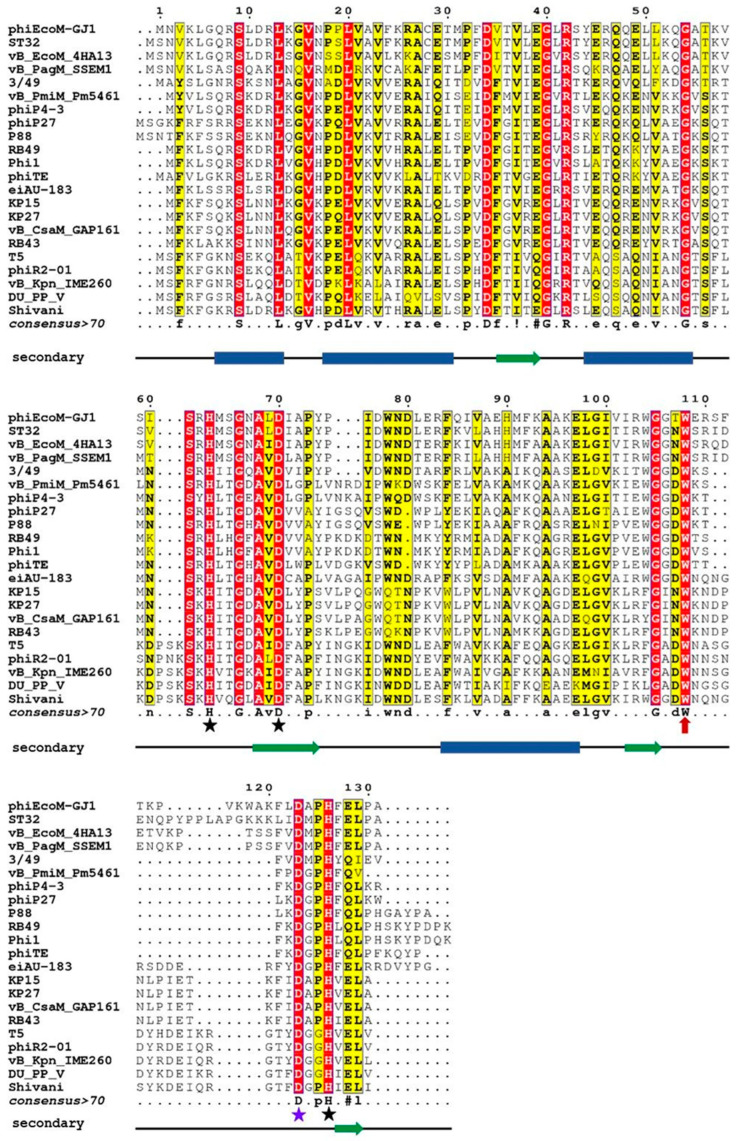
Multiple sequence alignment of amino acid sequences of orthologous peptidases of bacteriophages of the taxon *Caudovirales*. Conservative amino acids are colored red, similar residues are written with black bold characters and boxed in yellow; consensus is represented on a separate line. A consensus sequence is generated using criteria from MultAlin: uppercase is identity, lowercase is consensus level > 0.5, ! is any of IV, # is any of NDQE. The black stars mark the amino acids that coordinate the zinc atom, the violet star marks the catalytic aspartate. The line under consensus is the result of prediction of the secondary structure: blue rectangles mark α-helices, green arrows mark β-strands. Abbreviations denote endolysins of the following bacteriophages (in parentheses are protein numbers in GenBank): phiEcoM-GJ1—*Escherichia* phage phiEcoM-GJ1 (YP_001595416.1), ST32—*Escherichia* phage ST32 (YP_009790684.1), vB_EcoM_4HA13—*Escherichia* phage vB_EcoM_4HA13 (YP_009884078.1), vB_PagM_SSEM1—*Pantoea* phage vB_PagM_SSEM1 (YP_009859342.1), 3/49—*Shewanella* sp. phage 3/49 (YP_009103932.1), vB_PmiM_Pm5461—*Proteus* phage vB_PmiM_Pm5461 (YP_009195522.1), phiP4-3—*Proteus* phage phiP4-3 (YP_010093702.1), phiP27—*Enterobacteria* phage phiP27 (NP_543082.1), P88—*Escherichia* phage P88 (YP_009113074.1), RB49—*Enterobacteria* phage RB49 (NP_891673.1), Phil1—*Escherichia* phage Phi1 (YP_001469446.1), phiTE—*Pectobacterium* phage phiTE (YP_007392609.1), eiAU-183—*Edwardsiella* phage eiAU-183 (YP_009004687.1), KP15—*Klebsiella* phage KP15 (YP_003580002.1), KP27—*Klebsiella* phage KP27 (YP_007348788.1), vB_CsaM_GAP161—*Cronobacter* phage vB_CsaM_GAP161 (YP_006986425.1), RB43—*Enterobacteria* phage RB43 (YP_239135.1), T5—*Escherichia* phage T5 (AAS19387.1), phiR2-01—*Yersinia* phage phiR2-01 (YP_007237012.1), vB_Kpn_IME260—*Klebsiella* phage vB_Kpn_IME260 (YP_009597415.1), vB_PP_V—*Pectobacterium* phage DU_PP_V (YP_009795235.1), Shivani—*Salmonella* phage Shivani (YP_009194685.1).

**Figure 2 ijms-24-13249-f002:**
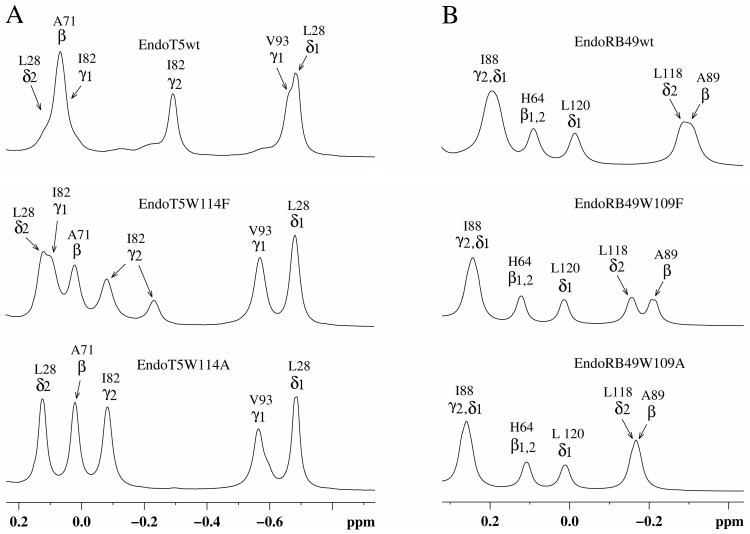
Upfield regions of aliphatic parts of ^1^H NMR spectra of EndoT5(Zn^2+^Ca^2+^) and EndoRB49(Zn^2+^) protein forms. (**A**) EndoT5wt and its mutants EndoT5W114F and EndoT5W114A; (**B**) EndoRB49wt and its mutants Endo RB49W109F and Endo RB49W109A.

**Figure 3 ijms-24-13249-f003:**
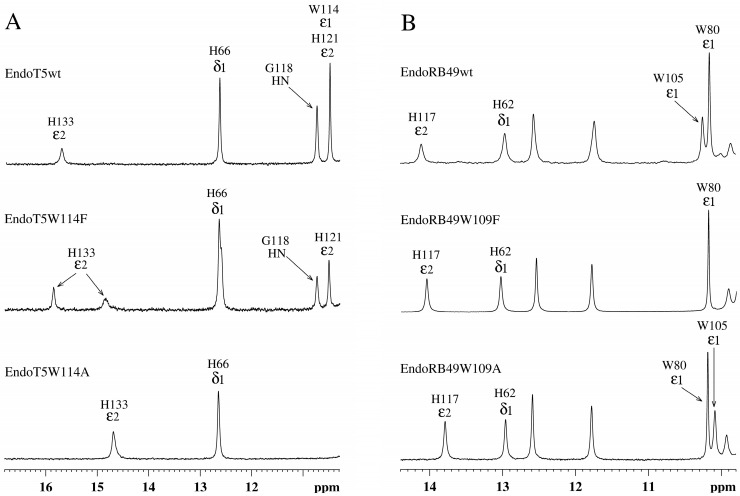
Downfield regions of aromatic parts of ^1^H NMR spectra of EndoT5(Zn^2+^Ca^2+^) and EndoRB49(Zn^2+^) protein forms. (**A**) EndoT5wt and its mutants EndoT5W114F and EndoT5W114A. (**B**) EndoRB49wt and its mutants Endo RB49W109F and Endo RB49W109A.

**Figure 4 ijms-24-13249-f004:**
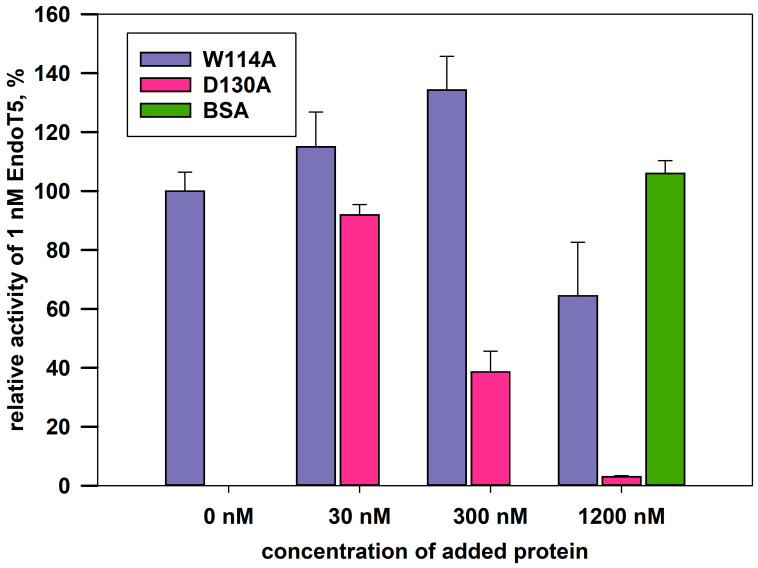
Effect of excess mutant proteins on the activity of 1 nM native EndoT5 under conditions of competition for the substrate. W114A, EndoT5 mutant for conservative tryptophan; D130A, EndoT5 mutant for catalytic aspartate; BSA, bovine serum albumin. Maximum EndoT5 activity in the absence of other proteins was taken as 100%.

**Figure 5 ijms-24-13249-f005:**
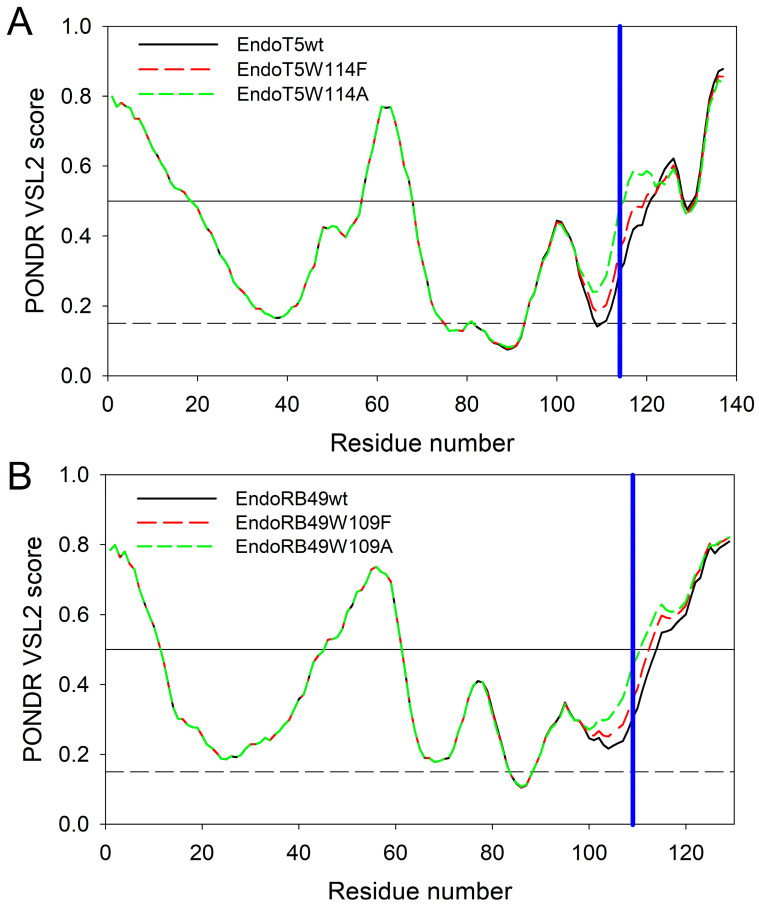
Intrinsic disorder profiles of EndoT5wt (black solid line), EndoT5W114F (red dashed line), and EndoT5W114A (green dashed line) (**A**) and EndoRB49wt (black solid line), EndoRB49W109F (red dashed line), EndoRB49W109A (green dashed line) (**B**) generated by PONDR^®^ VSL2. Thresholds for protein intrinsic disorder (disorder score 0.5) and conformational flexibility (0.15) are shown by solid and dashed black lines. Positions of conserved tryptophan residues W114 and W109 in EndoT5 and EndoRB49 are shown by vertical blue lines.

**Figure 6 ijms-24-13249-f006:**
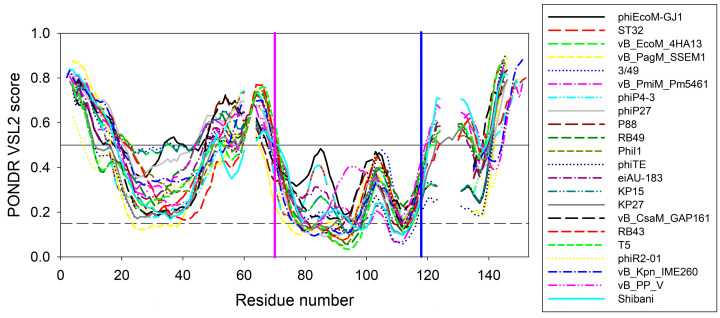
Aligned disorder profiles of 22 members of the phage l,d-peptidases of the M15_C family. Gaps in the disorder profiles correspond to gaps in the multiple sequence alignment shown in Figure 1. Vertical pink and blue lines indicate position of the zinc-coordinating aspartate residue and conserved tryptophan residue analyzed in this study, respectively.

**Table 1 ijms-24-13249-t001:** Comparison of the maximal specific lytic activities of EndoT5, EndoRB49 endolysins and their mutants for conservative tryptophan.

EndoT5 and Its Mutants		EndoRB49 and Its Mutants	
Protein	Specific Activity, U/mg	Protein	Specific Activity, U/mg
EndoT5wt	8380 ± 140	EndoRB49wt	3205 ± 134
EndoT5W114F *	2416 ± 25	EndoRB49W109F	1281 ± 102
EndoT5W114A *	-	EndoRB49W109A	151 ± 31

* Activity was measured in the presence of 100 μM CaCl_2_.

## Data Availability

PDB structure 8P3A. https://doi.org/10.2210/pdb8P3A/pdb, release date on 5 July 2023. The data present in the current study are available from the corresponding authors on reasonable request.

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
