# Peer review of "Conservative Tryptophan Residue in the Vicinity of an Active Site of the M15 Family l,d-Peptidases: A Key Element in the Catalysis"

_ijms, 2023, doi:10.3390/ijms241713249_

Round 1

Reviewer 1 Report

"The manuscript is well-written and includes important insights regarding the mechanism of functioning of L-alanoyl-D-glutamate peptidases of the M15_C family - peptidoglycan degraders in tailed phages.

The manuscript, authored by G. V. Mikoulinskaia and V. N. Uversky, contributes to the field of bioinformatics and antibacterial agents. The study focuses on globular single-domain endolysins from bacteriophages infecting Gram-negative bacteria. These endolysins have garnered attention as promising antibacterial agents for treating bacterial infections, including antibiotic-resistant strains, as well as having potential uses in biotechnology.

The central focus of the research revolves around a conserved tryptophan residue (Trp114/109) found near the active site of L,D-peptidases of the M15_C family. The authors embark on a thorough exploration of this residue's role by employing site-directed mutagenesis on two representative L,D-peptidases, EndoT5 and EndoRB49.
The results presented in the paper are convincing. The 1H NMR spectra analysis provided insights into the structural alterations induced by the mutagenesis. The observed changes in the hydrophobic core and active center of the protein highlight the residue's direct impact on the enzymatic function.

After careful evaluation, I would like to recommend the manuscript "Conservative tryptophan residue in the vicinity of an active site of the M15 family L,D-peptidases: A key element in the catalysis" for publication in IJMC as is."

Author Response

We are thankful to this reviewer for supportive comments. 

Reviewer 2 Report

L-alanoyl-D-glutamate peptidases EndoT5 and EndoRB49 of the M15 family were investigated. These enzymes (endolysins) might have future applications as broad spectrum antibiotics, therefore the topic is important.

The Authors continued their earlier work now aimed at determining the roles of conserved tryptophan residue W114/110/109 in interaction of the L,D-peptidases with the substrate and the effects on the enzymatic activity. To this end, site-directed mutagenesis was carried out based on bioinformatics analysis. They conclude, that conserved Trp114/109 position is a must to form the catalytically active “open” conformations of these proteins.

However, the manuscript can be improved according to the comments below:

Comments on manuscript:

151: Structural basis of functional changes

Limited experimental details on the NMR experimental part are given.

What was the basis of 1H-NMR signal assignment ?

No references to earlier NMR studies or assignments are mentioned.

Unlabelled proteins were used ?

Natural abundance 15N HSQC spectra (sofast) is possible to pick up ?

1024 scans for good quality 1H-NMR 1D spectra means that protein concentrations might have been very low.

164: Fig. 2

Please explain in more details the structural reasons of chemical shift movements,

perhaps a Figure on the structure of the hydrophobic core would be of help ?

Fig 3: aromatic and NH region, would be useful an overview of 1H spectra in the 5-16 ppm region (in supplementary material)

203: signal intensity loss is apparent for the eye, but quantitative description is not given

324: protein concentrations were determined, however the numerical values are missing

Other, technical issues:

Figure 1, red star is in fact violet

Some references have no doi.

Supplementary material is not provided for the reviewer

Author Response

151: Structural basis of functional changes

Limited experimental details on the NMR experimental part are given.

What was the basis of 1H-NMR signal assignment?

No references to earlier NMR studies or assignments are mentioned.

RESPONSE: Thank you for pointing this out. For clarification, we have added the following text at the beginning of the section 2.3 followed by references to the data:

Assignments of the Zn2+ and Zn2+Ca2+ forms of 13C15N-labelled EndoT5 were performed using a standard sets of 2D and 3D experiments and deposited to PDB (PDB ID: 2MXZ and 8P3A, correspondently). Assignment of the signals of histidine residues of the Zn2+ form of the orthologous EndoRB49 was performed using 1D-NOE with unlabeled protein and 15N HSQC experiments with 15N HSQC experiments with 13C15N-labelled protein published in [16]. Based on these data we analyzed the mutation-induced structural changes in the 1H NMR spectra.

Unlabelled proteins were used?

RESPONSE: Yes, we used unlabeled proteins.

Natural abundance 15N HSQC spectra (sofast) is possible to pick up?

RESPONSE: Assignment of EndoRB49 histidine residue signals was performed using 1D-NOE with unlabeled sample and 15N HSQC experiments with 13C15N-labelled sample published in [16]. Therefore, there was no need to obtain natural abundance sofast 15N HSQC spectra.

1024 scans for good quality 1H-NMR 1D spectra means that protein concentrations might have been very low.

RESPONSE: The protein concentration used at acquisition of 1H-NMR 1D spectra was 0.8 mM, which is significant. Multiple scans were required to obtain high-quality spectra was due to the low signal intensity of the labile HNε2 proton of the His 117/133 residue. We have added these protein concentration data to the section 3.3.

164: Fig. 2

Please explain in more details the structural reasons of chemical shift movements,

perhaps a Figure on the structure of the hydrophobic core would be of help?

RESPONSE: CYANA 2.1 program does not quantify the clustering of protein states within the obtained ensemble of structures. Therefore, we cannot reveal fine structural rearrangements inside the hydrophobic core. Changes in signal positions indicate only the fact of a change in the hydrophobic core packing.

Fig 3: aromatic and NH region, would be useful an overview of 1H spectra in the 5-16 ppm region (in supplementary material)

RESPONSE: The 5-9 ppm regions contain a lot of overlapping signals, which are difficult to interpret unambiguously. Figure 3 shows the most informative 9-16 ppm region with the characteristic differences due to mutations.

203: signal intensity loss is apparent for the eye, but quantitative description is not given

RESPONSE: For the EndoT5W114F mutant, there is a 63% reduction in signal intensity at -0.291 ppm. Thus, the population of molecules in the open conformation is 37%.

We have added these data to the section 2.4.

324: protein concentrations were determined, however the numerical values are missing

RESPONSE: The protein concentration used upon acquisition of 1H-NMR 1D spectra was 0.8 mM, we added these data to the section 3.3.

Other, technical issues:

Figure 1, red star is in fact violet

RESPONSE: Thank you for pointing this out. Wwe changed it in the caption to the picture.

Some references have no doi.

We added doi to reference 19:

DOI: 10.1039/c5ra05993c

Supplementary material is not provided for the reviewer

We have no supplementary material.